# Relationship between Dietary *n*-6 Fatty Acid Intake and Hypertension: Effect of Glycated Hemoglobin Levels

**DOI:** 10.3390/nu10121825

**Published:** 2018-11-24

**Authors:** Haruki Nakamura, Akinori Hara, Hiromasa Tsujiguchi, Thao Thi Thu Nguyen, Yasuhiro Kambayashi, Sakae Miyagi, Yohei Yamada, Keita Suzuki, Yukari Shimizu, Hiroyuki Nakamura

**Affiliations:** Department of Environmental and Preventive Medicine, Graduate School of Medical Science, Kanazawa University, 13-1 Takara-machi, Kanazawa, Ishikawa 920-8640, Japan; ahara@m-kanazawa.jp (A.H.); t-hiromasa@med.kanazawa-u.ac.jp (H.T.); toi_fs@yahoo.com (T.T.T.N.); ykamba@med.kanazawa-u.ac.jp (Y.K.); smiyagi@staff.kanazawa-u.ac.jp (S.M.); yamada503597@gmail.com (Y.Y.); keitasuzuk@yahoo.co.jp (K.S.); h_zu@me.com (Y.S.); hiro-n@po.incl.ne.jp (H.N.)

**Keywords:** blood pressure, hypertension, nutrition, *n*-6 fatty acid, population study, Japanese

## Abstract

The relationship between dietary *n*-6 fatty acids and hypertension is not clear. The metabolic products of *n*-6 fatty acids include those that control blood pressure, such as prostaglandin and thromboxane, and that differ depending on the extent of glucose tolerance. This cross-sectional study investigated the association of dietary *n*-6 fatty acid intake on hypertension, and the effects of glycated hemoglobin (HbA1c) value in 633 Japanese subjects aged 40 years and older. Dietary intake was measured using a validated brief self-administered diet history questionnaire. We defined hypertension as the use of antihypertensive medication or a blood pressure of 140/90 mmHg. The prevalence of hypertension was 55.3%. A high *n*-6 fatty acids intake inversely correlated with hypertension in subjects with HbA1c values less than 6.5% (odds ratio, 0.857; 95% confidence interval, 0.744 to 0.987). On the contrary, in subjects with an HbA1c value of 6.5% or higher, the *n*-6 fatty acids intake was significantly associated with hypertension (odds ratio, 3.618; 95% confidence interval, 1.019 to 12.84). Regular dietary *n*-6 fatty acid intake may contribute to the prevention and treatment of hypertension in a healthy general population. By contrast, in subjects with diabetes, regular *n*-6 fatty acids intake may increase the risk of hypertension.

## 1. Introduction

Prevention of hypertension is an important public health issue, and there is a direct positive relationship between hypertension and cardiovascular morbidity and mortality [1,2,3].

There is a relationship between a reduction in blood pressure (BP) and lifestyle choices, such as reducing sodium intake [4], weight loss [5], increased physical activity [6], and moderation in drinking [7]. High consumption of saturated fatty acids [8,9] and a low intake of polyunsaturated fatty acids (PUFA) [8,10] are related to a risk of hypertension. PUFA are classified into two series, *n*-3 fatty acids and *n*-6 fatty acids, depending on the position of the last double bond from the terminal methylcarbon. There is a beneficial effect of dietary *n*-3 fatty acid intake on cardiovascular disease prevention and a relationship between consumption of *n*-3 fatty acids and the prevention of hypertension [8,11].

There is a relationship between reduced levels of *n*-6 fatty acids, mainly linoleic acid (LA), and a higher risk of hypertension [8,12,13,14,15,16]. For example, the INTERMAP study, an international cross-sectional study of 4680 subjects, reported that dietary LA intake was inversely related to hypertension. The ARIC study, a longitudinal study designed to investigate the association of fatty acid composition of plasma cholesterol esters on the incidence of hypertension, found that higher LA was associated with a lower risk of incident hypertension. However, findings from other studies did not demonstrate an inverse association between the intake of *n*-6 fatty acids and hypertension [17,18]. The relationship between *n*-6 fatty acid intake and hypertension compared to the relationship between *n*-3 fatty acid intake and hypertension, remains to be elucidated.

The mechanism underlying the association between *n*-6 fatty acid intake and BP regulation is not fully understood. The metabolites of *n*-6 fatty acids, such prostaglandin (PG) and thromboxane (TX), are involved in BP regulation [19,20,21]. PGI2 and PGE2 are metabolites of PG produced by *n*-6 fatty acids, and they lower BP through vasodilation. However, PG metabolism varies by the extent of glucose tolerance, and the activation of TXA2 and TXA2/PGI2 ratio are elevated in diabetic state [22,23,24,25]. TXA2 has a strong vasoconstriction effect, and an elevated TXA2/PGI2 ratio is associated with arteriosclerosis [22,23,24,25].

The effect of dietary *n*-6 fatty acids on BP may differ in people according to glucose metabolism disorder. There are few studies that have investigated the relationship between the consumption of dietary *n*-6 fatty acids and hypertension in subjects with diabetes, and that have demonstrated its association according to the extent of glucose tolerance.

Here, we examined the association of dietary *n*-6 fatty acid intake on hypertension, and investigated the difference in this relationship according to a glycated hemoglobin (HbA1c) value.

## 2. Materials and Methods

### 2.1. Study Design and Participants

We investigated the relationship between dietary *n*-6 fatty acid intake and hypertension, and focused on the difference in this relationship according to a glycated hemoglobin (HbA1c) value. The study’s design was cross-sectional, and the present study was a part of the SHIKA study, a longitudinal community-based observational study that had been conducted among the residents of Shika town. Shika town is a country town in Ishikawa Prefecture of Japan and its population was 21,245 (15,088 residents were 40 years or older) in 2016 [26]. 

The target population of the present study was all middle-aged persons legally residing in a specific elementary school district (*n* = 2160), the Horimatsu district and the Higashimasuho district. The Shika Municipal Government supported the SHIKA study, and provided a list of all residents. The self-administered questionnaire and health examination data received between March 2014 and January 2016 in the SHIKA study was used. 

The recruitment of participants was shown in Figure 1. Questionnaires were mailed to 2160 participants, and 1987 participants returned them. Medical examinations were solicited from the 1987 participants, and active collaborators were included in this study. Among the 1987 participants, 837 participants were voluntary applicants for medical examinations, and 1150 participants did not undergo a medical examination. 47 subjects were excluded for lack of examination data, and 790 participants completed a medical examination and questionnaires. Subjects who were undergoing treatment of diabetes, dyslipidemia, coronary artery disease, or cerebrovascular disease were excluded because of the possibility of treatment bias of dietary guidance by physicians. Data from 633 individuals were finally analyzed.

### 2.2. Blood Pressure Measurements

BP was assessed twice in succession by well-trained nurses and clinical technologists. They had completed training courses at specialized medical check-up centers (ISHIKAWA HEALTH SERVICE ASSOCIATION, Ishikawa, Japan). BP was measured using automated digital sphygmomanometers based on the oscillometric method, UM-15P (Parama-tech Co., Ltd., Fukuoka, Japan) and HEM-907 (OMRON Co., Ltd., Kyoto, Japan), at rest in a sitting position. BP was calculated as the mean of two values and participants with a greater systolic blood pressure (SBP) than 140 mmHg, a diastolic blood pressure (DBP) than 90 mmHg, or the use of antihypertensive medication were considered to have hypertension.

### 2.3. Nutritional Assessment

Standardized methodology was used to estimate nutritional intake from data obtained in a Japanese version of the food-frequency questionnaire, a brief self-administered diet history questionnaire (BDHQ). BDHQ is a 4-page fixed questionnaire, and it asks subjects about the consumption frequency of selected 58 food and beverage items. The 58 items are selected from foods that are commonly consumed in Japan, and most of them are mainly from the food list used in the National Health and Nutrition Survey of Japan. The estimated dietary intake was calculated using an ad hoc computer algorithm, which included weighting factors for BDHQ [27].

The reproducibility and validity of BDHQ has been reported [27,28]. In a validation study by the BDHQ developers, the estimated energy and nutrient intakes calculated from the BDHQ among Japanese adults were compared with 16-day dietary records as reference. For many energy-adjusted nutrients, correlation values were greater than 0.4 between BDHQ and dietary record. Regarding *n*-6 fatty acids, the Pearson correlation coefficients with the dietary records for the energy-adjusted intakes ranged from 0.49 in men and from 0.55 in women [27]. The BDHQ has been considered to have a satisfactory ranking ability in the Japanese population.

### 2.4. Other Variables

Weight, height, waist circumference, and glycated hemoglobin (HbA1c) value were measured at health check-ups. We defined subjects with a HbA1c value of 6.5% or higher as a high HbA1c level. This definition of a high HbA1c level was the same criteria as that of The Japan Diabetes Society for diagnosing diabetes. Body mass index (BMI) was calculated as the weight in kilograms divided by the square of height in meters.

Self-administered questionnaires were used to assess other variables such as age, sex, smoking status, frequency of exercise, and drinking habits. Smoking status was classified into two groups: non-smokers and ex-smokers in one group, and current smokers in another group. Regarding drinking habit, we defined subjects who had drunk more than a glass of sake (22 g ethanol) per day three times a week or who had drunk at least four times a week as subjects with a drinking habit. For the frequency of exercise, subjects were classified into two groups. Subjects were asked whether they had exercised for more than 30 min at least two times a week for one year or not. They also asked whether they had performed tasks such as walking, cleaning, and carrying baggage for more than 1 h a day daily or not. When they replied in the affirmative to either of these questions, they were considered to have an exercise habit.

### 2.5. Statistical Analysis

Nutritional intake was adjusted for energy using the density method as a percentage of the daily energy intake for energy-containing nutrients. All subjects were allocated to a BP group (hypertension group or normal BP group) and an HbA1c group (high HbA1c level or normal HbA1c level). Descriptive statistics of basic characteristics among different BP groups were compared. A Student’s *t*-test was used to compare the average of continuous variables, and a chi-squared test was used to compare the proportions of categorical variables. To examine the interaction in dietary *n*-6 fatty acid intake between BP and HbA1c groups, a two-way analysis of variance (two-way ANOVA) was used. The relationship between BP and nutrient intake was examined by multiple logistic regression analysis. There are two models with different adjustments for confounding factors. Model 1 adjusted for sex (male subjects or female subjects), age (continuous), BMI (continuous), frequency of exercise (yes or no), and smoking status (non-smokers and ex-smokers or current smokers); Model 2 adjusted for all variables in the model 1, plus drinking habits (yes or no) and sodium intake (continuous).

In all analyses, the *p* values presented were two-sided, with the statistical significance being determined by a false discovery rate of less than 0.05. The IBM SPSS Statistics version 24.0 for Mac (SPSS Inc., Armonk, NY, USA) were used for all analysis.

### 2.6. Ethics Statement

Informed consent was obtained from all participants. All authors had full access to all of the data and consented to publication. The SHIKA study was approved by the Ethical Committee at Kanazawa University.

## 3. Results

### 3.1. Participant Characteristics in Different Blood Pressure Groups

Table 1 illustrates the basic characteristics and nutrients intake. The mean (standard deviation, SD) age of participants was 61.0 (11.5) years. 47% were male, 21% were current smokers, 42% were drinkers, and 55% had a habit of exercise. The mean (SD) SBP and DBP of all subjects were 139 (20.4) mmHg and 80.7 (11.8) mmHg, respectively. Regarding differences of gender, female subjects had a lower BMI, SBP, and DBP, and they consumed more *n*-6 fatty acid and LA. There was no significant difference in HbA1c between male and female subjects.

When the subjects were classified into two groups according to the definition of hypertension, 350 subjects satisfied the criteria of hypertension; 184 subjects were on medication for hypertension. Ratios of male subjects (*p* < 0.001), age (*p* < 0.001), the number of drinkers (*p* < 0.001) and the consumption of sodium (*p* = 0.024) were higher in the hypertension group than in normal BP group (Table 2). The consumption of *n*-6 fatty acids (*p* < 0.001) and LA (*p* < 0.001) was significantly higher in the normal BP group than in the hypertension group. There was no difference in smoking habits, the frequency of exercise, and BMI and the consumption of *n*-3 fatty acids between the BP groups.

### 3.2. Interaction of HbA1c Level

Figure 2 demonstrated the spread of relationship between the dietary LA intake and the blood levels of HbA1c. In subjects with normal HbA1c levels, Figure 2A,B suggested that the number of subjects consuming high LA was more in normal BP subjects than in hypertensive subjects. In subjects with high HbA1c levels, there was a tendency for hypertensive subjects to consume high intake of LA, compared with normal BP subjects.

In Table 3, two-way ANOVA demonstrated that there was a significant interaction between BP groups and HbA1c groups on the total intake of *n*-6 fatty acids (*p* = 0.035) and LA (*p* = 0.033). This result suggests that there is a relationship between the intake of *n*-6 fatty acids and hypertension, which depends on the HbA1c level. For other *n*-6 fatty acids such as such as gamma-linolenic acid, eicosadienoic acid, dihomo-gamma-linolenic acid, arachidonic acid, and docosapentaenoic acid (DPA), no significant interactions were observed. Unlike the interaction of HbA1c groups, there were no significant interaction between BP groups and gender groups in the consumption of *n*-6 fatty acids (Appendix A)

### 3.3. Characteristics of the Study Population in Different Blood Pressure Groups according to HbA1c Level

Characteristics of the study population in different blood pressure groups according to HbA1c level are presented in Table 4. In subjects with normal HbA1c level, the ratios of male subjects (*p* < 0.001), age (*p* < 0.001), number of drinkers (*p* < 0.001), BMI (*p* < 0.001), and sodium intake (*p* = 0.030) were higher in the hypertension group than in the normal BP group. There was a significant difference in *n*-6 fatty acids and LA intake among BP groups; normal BP groups consumed more *n*-6 fatty acids (*p* < 0.001) and LA (*p* < 0.001) than the hypertension group. 

The consumption of *n*-6 fatty acids and LA were higher in the hypertension groups than in the normal BP groups in subjects with high HbA1c levels, although the difference was not statistically significant in Student’s *t*-tests.

### 3.4. Relationship between n-6 Fatty Acid Intake and Hypertension according to HbA1c level.

Table 5 shows the relationship between BP and the consumption of *n*-6 fatty acids using multiple logistic regression analysis. Following the results of Table 2, which demonstrated the interaction between the BP groups and HbA1c groups, subjects were stratified basis on HbA1c levels. High consumption of *n*-6 fatty acids were inversely correlated with hypertension in subjects with a normal HbA1c level after adjustment confounding factors; the *p* values for total *n*-6 fatty acids and LA were 0.032 (odds ratio (OR) = 0.857; 95% confidence interval (CI) = 0.744 to 0.987) and 0.037 (OR = 0.858; 95% CI = 0.744 to 0.991), respectively. 

On the other hand, in subjects with a high HbA1c level, high consumption of total *n*-6 fatty acids (OR = 3.618; 95% CI = 1.019 to 12.84) and LA (OR = 3.986; 95% CI = 1.050 to 15.13) were significantly associated with hypertension after adjustment confounding factors.

## 4. Discussion

We found that the relationship between dietary *n*-6 fatty acid intake and hypertension was different, depending on the difference in HbA1c level: a high intake of *n*-6 fatty acids was significantly associated with hypertension in subjects with an HbA1c value of 6.5% or higher, but a high intake of *n*-6 fatty acids was inversely associated with hypertension in subjects with a HbA1c values less than 6.5%. 

Fatty acids are divided into three groups, saturated fatty acids, monounsaturated fatty acids, and PUFA, according to the degree of unsaturation. An increased consumption of saturated fatty acids is a risk for hypertension [8,9]. Previous studies reported an inverse relationship between monounsaturated fatty acids and cardiovascular mortality [29,30], but there is little evidence about the relationship between monounsaturated fatty acid intake and BP. For PUFA, the consumption of omega-3 fatty acids from marine oils has a hypotensive effect, and is beneficial for preventing cardiovascular disease [11,31,32]. Elevated consumption of omega-3 polyunsaturated fatty acids reduces BP in hypertensive subjects [8,33]. In the present study, because strong positive correlations were suggested between *n*-6 fatty acid intake and saturated fatty acid intake, and between *n*-6 fatty acid intake and *n*-3 fatty acid (data not shown), we did not adjust consumption of them in multiple logistic regression analysis. 

We found that the consumption of *n*-6 fatty acids was inversely correlated with hypertension in all subjects, and in subjects a with a HbA1c values less than 6.5% (Table 4). These results were consistent with previous studies that demonstrated that a high tissue level of *n*-6 fatty acids, especially LA, decreases BP [12,13,14,15,16]. One such example is the INTERMAP study, which was an international cross-sectional study that was conducted to survey the relationship between LA intake and hypertension in 4680 men and women ages 40–59 years from 17 population samples in Japan, China, the United Kingdom, and the United States. The INTERMAP study reported that a higher intake of dietary LA was associated with a decrease in BP, and an inverse relationship was stronger in non-intervened persons (not on a special diet, not consuming nutritional supplements, not diagnosed with cardiovascular disease or diabetes, and not taking medication for high BP, cardiovascular disease, or diabetes) [12]. The characteristics of the subjects in the present study were similar to those in the INTERMAP study, because we excluded subjects who were undergoing treatment for diabetes, dyslipidemia, coronary artery disease, and cerebrovascular disease, to exclude the effects of medications and lifestyle guidance by physicians. The ARIC study also supported our results: The ARIC study was a longitudinal study that was designed to investigate the association of the fatty acid composition of plasma cholesterol esters on the incidence of hypertension, and demonstrated that LA was negatively associated with a high risk of hypertension after adjustment for confounding factors, such as age, sex, BMI, waist/hip ratio, smoking status, ethanol intake, education level, sport index, and baseline SBP [8]. Our results were consistent with the previous findings, and they suggest that a regular intake of *n*-6 fatty acids may prevent hypertension in a general population.

We found that the relationship between the intake of *n*-6 fatty acids and hypertension was different, depending on the difference in HbA1c level (Table 2) and that a high consumption of *n*-6 fatty acids was significantly related to hypertension in subjects with an HbA1c value of 6.5% or higher (Table 4). To the best of our knowledge, the relationship between the consumption of dietary *n*-6 fatty acids and hypertension by the extent of a HbA1c level has not yet been studied. This is the first study that demonstrates an interaction between BP and a HbA1c level with regard to the dietary consumption of *n*-6 fatty acids. 

The effects of BP on *n*-6 fatty acid intake is mediated through changes in PG metabolism. The production of PG and TX affect vasodilation and platelet aggregation, and it is involved in BP regulation [19,20,21]; *n*-6 fatty acids may lower BP by increasing PG vasodilators such as PGI2 and PGE2 [19,20]. In addition, PG metabolism in non-diabetic and diabetic subjects is different; the activation of TXA2 and the TXA2/PGI2 ratio are elevated in subjects with diabetes [22,23,24,25]. TXA2 has a strong effect on vasoconstriction, and an elevated TXA2/PGI2 ratio is associated with arteriosclerosis [22,23,24,25]. Therefore, differences in BP from *n*-6 fatty acid intake according to the HbA1c level are plausible, because PG metabolism varies by the extent of glucose metabolic disorder. We consider that when dietary *n*-6 fatty acids are converted into arachidonic acids in vivo, PG vasodilators such as PGI2 and PGE2 are produced from arachidonic acids, and then these vasodilator PGs lower BP in the general population. On the other hand, more vasoconstriction TXs such as TXA2 are produced from arachidonic acids in diabetic subjects than in non-diabetic subjects, and the elevated TXA2/PGI2 consequently causes hypertension in diabetic subjects. In addition, the elevated activation of TXA2 in the diabetic state is considered to cause hypertension.

We showed that a high consumption of *n*-6 fatty acids was significantly related to hypertension in subjects with an HbA1c value of 6.5% or higher. However, *n*-6 fatty acids may not always be beneficial for cardiovascular disease. Previous research has examined the efficacy of replacing saturated fatty acids with *n*-6 fatty acids, especially LA, for the secondary prevention of coronary artery disease, and showed that a higher intake of an LA-rich diet was significantly associated with a higher risk of death from all causes, cardiovascular disease, and coronary heart disease [34]. In the LA-rich group, 87% of participants had a history of myocardial infraction, and 13% had acute angina or coronary insufficiency; the prevalence of impaired glucose metabolism was approximately 30%. Our results support the possibility that *n*-6 fatty acid intake has no cardiovascular benefit in subjects who are at risk for cardiovascular disease.

There were several limitations in the present study. First, causality was not examined because the study had a cross-sectional design. To infer causality, a proper longitudinal study is needed. Second, the dietary data were obtained from BDHQ; BDHQ has a limited number of food and beverage items, and does not provide an accurate estimate of absolute intake. In addition, recall bias may affect reported dietary data. Although the validity of BDHQ is well established in Japan, dietary data should still be carefully interpreted. Third, a selection bias and a treatment effect bias needs to be considered. Participants were voluntary collaborators and they might be people who were concerned about being healthy. In addition, patients receiving medication for hypertension may also have been receiving lifestyle guidance from their physician. Fourth, data on serum levels of *n*-6 fatty acids were not used. *n*-6 fatty acids are essential fatty acids, and humans must ensure their intake because the body cannot synthesize them. With this taken into consideration, serum levels of *n*-6 fatty acids depend on the amount of dietary intake. However, the serum concentration of *n*-6 fatty acids is considered to be a better indicator of *n*-6 fatty acid status, and the relationship between serum levels of *n*-6 fatty acids and hypertension needs to be confirmed in further studies.

## 5. Conclusions

This cross-sectional study of a Japanese population from the SHIKA study examined the relationship between dietary *n*-6 fatty acid intake and hypertension. A high dietary consumption of *n*-6 fatty acids was significantly associated with hypertension in subjects with an HbA1c value of 6.5% or higher, but dietary consumption of *n*-6 fatty acids was inversely associated with hypertension in subjects with HbA1c values of less than 6.5%. The regular intake of dietary *n*-6 fatty acids may contribute to the prevention and treatment of hypertension in a generally healthy population. However, subjects with glucose metabolism disorder may not have this favorable effect, and they may have an increased risk of hypertension from the regular intake of dietary *n*-6 fatty acids.

## Figures and Tables

**Figure 1 nutrients-10-01825-f001:**
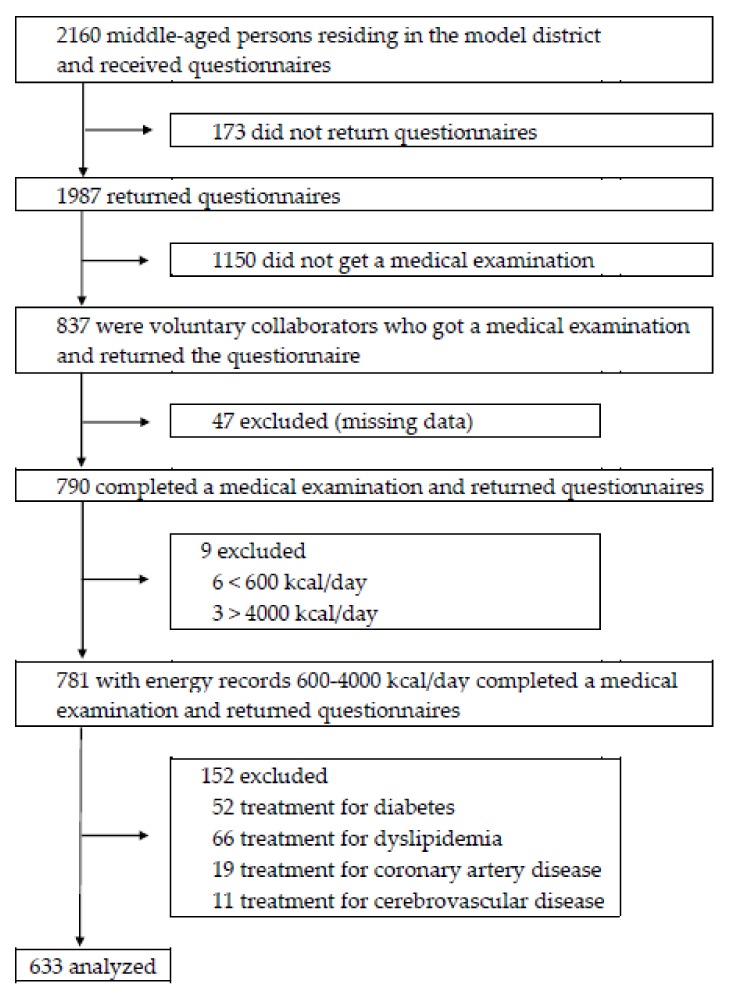
Flow chart.

**Figure 2 nutrients-10-01825-f002:**
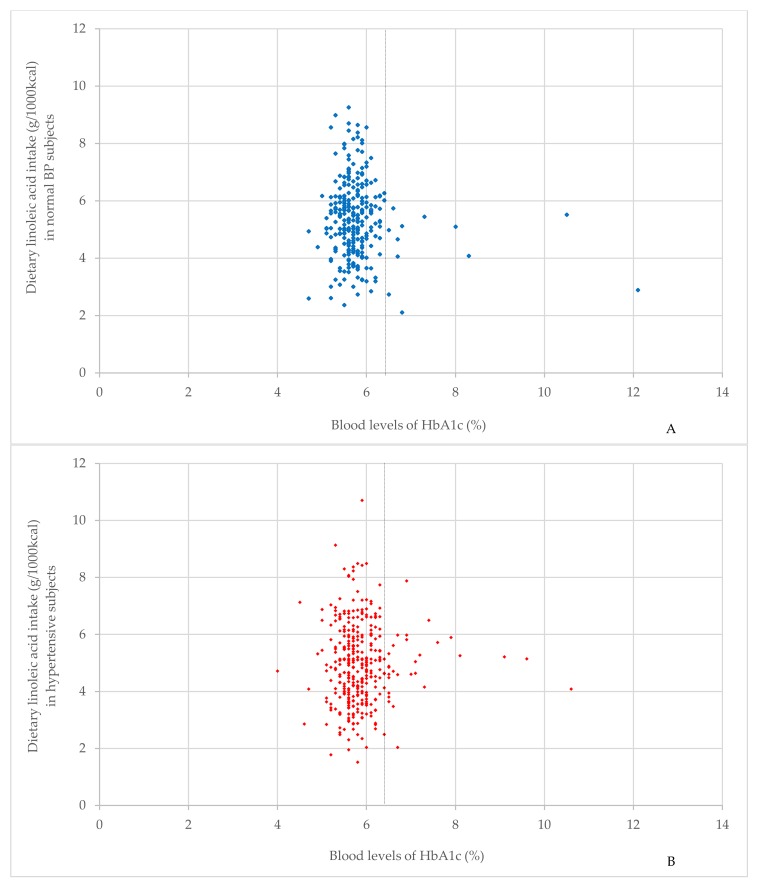
The spread of relationship between dietary linoleic acid intake and blood levels of HbA1c. (**A**) shows the spread of the relationship between dietary linoleic acid intake and blood levels of HbA1c in normal BP subjects (*n* = 283). (**B**) shows the spread of the relationship between dietary linoleic acid intake and blood levels of HbA1c in hypertensive subjects (*n* = 350). The vertical dotted line indicates the cut-off for the normal HbA1c level (HbA1c = 6.5). Hypertensive subjects were defined as participants with a greater SBP than 140 mmHg, a DBP greater than 90 mmHg, or the use of antihypertensive medication. A high HbA1c level was defined as an HbA1c value of 6.5% or higher. Abbreviations: BP, blood pressure; SBP, systolic blood pressure; DBP, diastolic blood pressure; HbA1c, hemoglobin A1c.

**Table 1 nutrients-10-01825-t001:** Participant characteristics in different blood pressure groups.

Characteristic	All Subjects	Male	Female	*p* Value
No. of subjects	633	299	334	
Age	61.0 (11.5)	60.8 (11.1)	61.2 (11.8)	0.733
Smoking status, *n* (%)				<0.001
Never or Ex-smoker	503 (79)	196 (66)	307 (92)	
Current	130 (21)	103 (34)	27 (8)	
Drinking habit, *n* (%)				<0.001
yes	268 (42)	206 (79)	62 (20)	
no	365 (58)	93 (31)	268 (80)	
Exercise habit, *n* (%)				0.526
yes	345 (55)	159 (53)	186 (57)	
no	288 (45)	140 (47)	148 (43)	
Height (cm)	160 (9.65)	168 (6.92)	154 (6.74)	<0.001
Weight (kg)	59.9 (11.8)	67.3 (10.4)	53.2 (8.39)	<0.001
Waist circumference (cm)	83.8 (9.26)	85.8 (8.70)	81.9 (9.37)	<0.001
BMI (kg/m^2^)	23.2 (3.24)	23.9 (2.98)	22.5 (3.32)	<0.001
SBP (mmHg)	139 (20.4)	143 (20.3)	135 (19.6)	<0.001
DBP (mmHg)	80.7 (11.8)	83.5 (12.0)	78.3 (11.0)	<0.001
HbA1c (NSGP) (%)	5.82 (0.605)	5.81 (0.549)	5.83 (0.383)	0.686
Energy and Nutrients				
Energy (kcal)	1863 (616)	2088 (630)	1611 (529)	<0.001
Nutrients (g/1000 kcal)				
Protein	38.1 (8.18)	14.4 (3.13)	38.3 (7.58)	0.637
Carbohydrate	134.1 (21.1)	134 (22.2)	134 (19.6)	<0.001
Sodium	2.45 (0.540)	2.40 (0.515)	2.49 (0.559)	0.036
Potassium	1.39 (0.430)	1.22 (0.349)	1.55 (0.467)	<0.001
Calcium	0.294 (0.113)	0.258 (0.103)	0.326 (0.112)	<0.001
Total dietary fiber	6.54 (2.19)	5.65 (1.75)	7.33 (2.25)	<0.001
SFA	7.27 (2.17)	6.48 (1.97)	7.97 (2.11)	<0.001
MUFA	9.76 (2.65)	8.95 (2.60)	10.5 (2.49)	<0.001
PUFA	6.82 (1.74)	6.33 (1.69)	7.26 (1.69)	<0.001
*n*-3 fatty acid	1.48 (0.503)	1.40 (0.474)	1.57 (0.516)	<0.001
*n*-6 fatty acid	5.30 (1.41)	4.91 (1.36)	5.67 (1.35)	<0.001
LA	5.15 (1.38)	4.76 (1.33)	5.50 (1.33)	<0.001
GLA (mg/1000 kcal)	4.09 (2.68)	3.82 (2.44)	4.32 (2.87)	0.019
EA (mg/1000 kcal)	24.6 (8.54)	23.1 (8.33)	25.9 (8.52)	<0.001
DGLA (mg/1000 kcal)	16.0 (5.10)	14.8 (4.98)	17.0 (4.98)	<0.001
AA (mg/1000 kcal)	92.7 (30.7)	87.9 (31.6)	100 (29.4)	<0.001
DPA (mg/1000 kcal)	5.25 (3.26)	5.07 (3.09)	5.42 (3.40)	0.175

Continuous variables are presented as means (SD). For continuous variables, we used Student’s *t*-tests to examine gender differences; for categorical variables, we used a chi-squared test to examine gender differences. Hypertensive subjects were defined as participants with a greater SBP than 140 mmHg, a DBP than 90 mmHg, or the use of antihypertensive medication. Abbreviations: BP, blood pressure; BMI, body mass index; SBP, systolic blood pressure; DBP, diastolic blood pressure; HbA1c, hemoglobin A1c; SFA, Saturated fatty acid; MUFA, Monounsaturated fatty acid; PUFA, Polyunsaturated fatty acid; LA, linoleic acid; GLA. Gamma-linolenic acid; EA, eicosadienoic acid; DGLA, dihomo-gamma-linolenic acid; AA, arachidonic acid; and DPA, docosapentaenoic acid.

**Table 2 nutrients-10-01825-t002:** Participant characteristics in different blood pressure groups.

Characteristic	Hypertension	Normal BP	*p* Value
No. of subjects	350	283	
Men, *n* (%)	196 (56)	103 (36)	<0.001
Age	64.5 (10.7)	56.7 (11.0)	<0.001
Smoking status, *n* (%)			0.311
Never or Ex-smoker	273 (78)	230 (81)	
Current	77 (22)	53 (19)	
Drinking habit, *n* (%)			<0.001
yes	178 (51)	90 (32)	
no	172 (49)	193 (68)	
Exercise habit, *n* (%)			0.496
yes	195 (56)	150 (53)	
no	155 (44)	133 (47)	
Height (cm)	160 (9.79)	160 (9.48)	0.233
Weight (kg)	61.4 (11.5)	58.0 (11.9)	0.419
Waist circumference (cm)	85.5 (9.02)	81.6 (9.13)	0.755
BMI (kg/m^2^)	23.8 (3.17)	22.4 (3.18)	0.755
SBP (mmHg)	151 (18.3)	124 (9.10)	<0.001
DBP (mmHg)	85.7 (12.2)	74.5 (7.44)	<0.001
HbA1c (NSGP) (%)	5.85 (0.593)	5.78 (0.617)	0.134
Energy and Nutrients			
Energy (kcal)	1895 (625)	1822 (604)	0.142
Nutrients (g/1000 kcal)			
Protein	38.0 (8.64)	38.3 (7.58)	0.637
Carbohydrate	134 (22.2)	134 (19.6)	0.933
Sodium	2.49 (0.540)	2.39 (0.536)	0.024
Potassium	1.39 (0.454)	1.39 (0.400)	0.934
Calcium	0.294 (0.114)	0.294 (0.111)	0.955
Total dietary fiber	6.59 (2.29)	6.48 (2.07)	0.528
SFA	6.87 (2.06)	7.76 (2.21)	<0.001
MUFA	9.33 (2.69)	10.3 (2.51)	<0.001
PUFA	6.62 (1.76)	7.06 (1.69)	0.002
*n*-3 fatty acid	1.48 (0.504)	1.49 (0.503)	0.728
*n*-6 fatty acid	5.12 (1.43)	5.54 (1.34)	<0.001
LA	4.96 (1.40)	5.38 (1.32)	<0.001
GLA (mg/1000 kcal)	3.94 (2.41)	4.26 (2.98)	0.151
EA (mg/1000 kcal)	23.7 (8.60)	25.6 (8.36)	0.005
DGLA (mg/1000 kcal)	15.4 (5.22)	16.6 (4.88)	0.003
AA (mg/1000 kcal)	90.4 (31.8)	95.5 (29.1)	0.037
DPA (mg/1000 kcal)	5.35 (3.20)	5.14 (3.34)	0.437

Continuous variables are presented as mean (SD). For continuous variables, we used Student’s *t*-tests; for categorical variables, we used a chi-squared test. Hypertensive subjects were defined as participants with a greater SBP than 140 mmHg, a DBP than 90 mmHg, or the use of antihypertensive medication. Abbreviations: BP, blood pressure; BMI, body mass index; SBP, systolic blood pressure; DBP, diastolic blood pressure; HbA1c, hemoglobin A1c; SFA, Saturated fatty acid; MUFA, Monounsaturated fatty acid; PUFA, Polyunsaturated fatty acid; LA, linoleic acid; GLA. Gamma-linolenic acid; EA, eicosadienoic acid; DGLA, dihomo-gamma-linolenic acid; AA, arachidonic acid; and DPA, docosapentaenoic acid.

**Table 3 nutrients-10-01825-t003:** Interaction between BP and HbA1c groups by *n*-6 fatty acid intake.

*n*-6 Fatty Acid	HbA1c Level	Hypertension	Normal BP	*p* Value for Interaction
		Average (SD)	Average (SD)	
*n*-6 fatty acid (g/1000 kcal)	Normal	5.12 (1.31)	5.59 (1.33)	0.035
	High	5.11 (1.10)	4.54 (1.22)	
LA (g/1000 kcal)	Normal	4.96 (1.43)	5.42 (1.31)	0.033
	High	4.94 (1.09)	4.37 (1.21)	
GLA (mg/1000 kcal)	Normal	3.87 (2.41)	4.21 (3.00)	0.682
	High	4.78 (2.25)	5.50 (2.52)	
EA (mg/1000 kcal)	Normal	23.6 (8.74)	25.6 (8.43)	0.949
	High	24.7 (6.95)	26.9 (6.90)	
DGLA (mg/1000 kcal)	Normal	15.3 (5.22)	16.6 (4.94)	0.585
	High	16.3 (5.27)	16.6 (3.57)	
AA (mg/1000 kcal)	Normal	89.9 (31.5)	95.4 (29.3)	0.629
	High	96.6 (34.9)	96.9 (27.5)	
DPA (mg/1000 kcal)	Normal	5.30 (3.14)	5.05 (3.33)	0.147
	High	5.87 (3.82)	7.29 (2.82)	

*p* values from a two-way analysis of variance means interaction between HbA1c groups and BP groups. Hypertensive subjects were defined as participants with a greater SBP than 140 mmHg, a DBP greater than 90 mmHg, or the use of antihypertensive medication. A high HbA1c level was defined as an HbA1c value of 6.5% or higher. Forty-one participants had a high HbA1c level; 12 high HbA1c level participants had a normal BP, and 29 had hypertension. There were 592 participants with a normal HbAc1 level; 271 had a normal BP, and 321 had hypertension. Abbreviations: BP, blood pressure; HbA1c, hemoglobin A1c; LA, linoleic acid; GLA. Gamma-linolenic acid; EA, eicosadienoic acid; DGLA, dihomo-gamma-linolenic acid; AA, arachidonic acid; and DPA, docosapentaenoic acid.

**Table 4 nutrients-10-01825-t004:** Participant characteristics in different blood pressure groups according to HbA1c level.

Characteristic	HbA1c < 6.5		*p*	HbA1c ≥ 6.5		*p*
	HTN	NBP		HTN	NBP	
No. of subjects	321	271		29	12	
Men, *n* (%)	178 (55)	96 (35)	<0.001	18 (62)	7 (58)	0.823
Age	64.2 (10.6)	56.5 (11.0)	<0.001	67.6 (11.6)	62.2 (9.47)	0.164
Smoking status, *n* (%)			0.533			0.254
Never or Ex-smoker	254 (79)	220 (81)		19 (66)	10 (83)	
Current	67 (21)	51 (19)		10 (34)	2 (17)	
Drinking habit, *n* (%)			<0.001			0.558
yes	163 (51)	85 (31)		15 (52)	5 (42)	
no	158 (49)	186 (69)		14 (48)	7 (58)	
Exercise habit, *n* (%)			0.261			0.141
yes	183 (57)	142 (52)		17 (59)	4 (33)	
no	138 (43)	129 (48)		12 (41)	8 (67)	
Height (cm)	160 (9.76)	160 (9.48)	0.951	161 (10.3)	162 (9.73)	0.661
Weight (kg)	61.1 (11.2)	57.6 (11.6)	<0.001	64.9 (14.0)	65.5 (15.8)	0.907
Waist circumference (cm)	85.1 (8.87)	81.4 (8.93)	<0.001	89.4 (9.84)	86.4 (12.5)	0.421
BMI (kg/m^2^)	23.7 (3.13)	22.3 (3.01)	<0.001	24.8 (3.50)	24.8 (5.43)	0.979
SBP (mmHg)	152 (18.5)	124 (9.23)	<0.001	148 (16.1)	124 (5.45)	<0.001
DBP (mmHg)	86.1 (12.2)	74.5 (7.46)	<0.001	81.8 (12.0)	74.0 (7.44)	0.016
HbA1c (NSGP) (%)	5.73 (0.342)	5.69 (0.299)	0.157	7.19 (1.01)	7.73 (1.80)	0.227
Energy and Nutrients						
Energy (kcal)	1905 (632)	1808 (590)	0.055	1788 (541)	2170 (823)	0.159
Nutrients (g/1000 kcal)						
Protein	37.9 (8.65)	38.3 (7.63)	0.578	39.2 (8.56)	39.2 (6.64)	0.992
Carbohydrate	134 (22.6)	134 (19.6)	0.947	134 (18.1)	135 (21.7)	0.904
Sodium	2.49 (0.545)	2.39 (0.535)	0.030	2.49 (0.496)	2.38 (0.576)	0.544
Potassium	1.39 (0.461)	1.39 (0.402)	0.852	1.43 (0.374)	1.38 (0.346)	0.733
Calcium	0.293 (0.116)	0.294 (0.116)	0.875	0.302 (0.096)	0.290 (0.098)	0.714
Total dietary fiber	6.57 (2.30)	6.49 (2.08)	0.638	6.74 (2.21)	6.21 (2.00)	0.482
SFA	6.84 (2.07)	7.79 (2.25)	<0.001	7.11 (1.99)	7.29 (0.867)	0.695
MUFA	9.33 (2.74)	10.3 (2.52)	<0.001	9.36 (2.12)	9.30 (1.33)	0.931
PUFA	6.62 (1.80)	7.11 (1.68)	0.001	6.70 (1.33)	6.11 (1.51)	0.220
*n*-3 fatty acid	1.47 (0.504)	1.49 (0.506)	0.632	1.57 (0.497)	1.54 (0.439)	0.852
*n*-6 fatty acid	5.12 (1.46)	5.59 (1.33)	<0.001	5.11 (1.10)	4.54 (1.22)	0.155
LA	4.96 (1.43)	5.42 (1.31)	<0.001	4.94 (1.09)	4.37 (1.21)	0.148
GLA (mg/1000kcal)	3.87 (2.41)	4.21 (3.00)	0.139	4.78 (2.25)	5.50 (2.52)	0.400
EA (mg/1000kcal)	23.6 (8.74)	25.6 (8.43)	0.006	24.7 (6.95)	26.9 (6.90)	0.372
DGLA (mg/1000kcal)	15.3 (5.22)	16.6 (4.94)	0.002	16.3 (5.27)	16.6 (3.57)	0.825
AA (mg/1000kcal)	89.9 (31.5)	95.4 (29.3)	0.027	96.6 (34.9)	96.9 (27.5)	0.975
DPA (mg/1000kcal)	5.30 (3.14)	5.05 (3.33)	0.349	5.87 (3.82)	7.29 (2.82)	0.201

Continuous variables are presented as mean (SD). For continuous variables, we used Student’s t-tests; for categorical variables, we used a chi-squared test. Hypertensive subjects were defined as participants with a greater SBP than 140 mmHg, a DBP than 90 mmHg, or the use of antihypertensive medication. Abbreviations: *p*, *p* value; BP, blood pressure; HTN, hypertension; NBP, normal blood pressure; BMI, body mass index; SBP, systolic blood pressure; DBP, diastolic blood pressure; HbA1c, hemoglobin A1c; SFA, Saturated fatty acid; MUFA, Monounsaturated fatty acid; PUFA, Polyunsaturated fatty acid; LA, linoleic acid; GLA. Gamma-linolenic acid; EA, eicosadienoic acid; DGLA, dihomo-gamma-linolenic acid; AA, arachidonic acid; and DPA, docosapentaenoic acid.

**Table 5 nutrients-10-01825-t005:** Association between *n*-6 fatty acid intake and hypertension by HbA1c groups.

HbA1c Level		Model 1	Model 2
		OR (95% CI, *p* Value)	OR (95% CI, *p* Value)
All subjects	*n*-6	0.898 (0.788 to 1.022, 0.104)	0.884 (0.771 to 1.013, 0.077)
	LA	0.899 (0.787 to 1.026, 0.114)	0.886 (0.771 to 1.019, 0.089)
High HbA1c	*n*-6	3.676 (1.060 to 12.76, 0.040)	3.618 (1.019 to 12.84, 0.047)
	LA	3.993 (1.090 to 14.63, 0.037)	3.986 (1.050 to 15.13, 0.042)
Normal HbA1c	*n*-6	0.870 (0.761 to 0.995, 0.041)	0.857 (0.744 to 0.987, 0.032)
	LA	0.885 (0.759 to 0.997, 0.045)	0.858 (0.744 to 0.991, 0.037)

Model 1: *p* values from a multiple logistic regression analysis after adjustments for the following independent factors: sex (male subjects or female subjects), age (continuous), BMI (continuous), frequency of exercise (yes or no) and smoking status (non-smokers and ex-smokers or current smokers). Model 2: *p* values by a multiple logistic regression analysis after adjustments for all variables in the Model 1, plus drinking habits and consumption of sodium; sex (male or female), age (continuous), BMI (continuous), frequency of exercise (yes or no), smoking status (non-smokers and ex-smokers or current smokers), drinking habits (yes or no), and sodium intake (continuous). Forty-one participants had a high HbA1c level; 592 participants had a normal HbAc1 level. Hypertension subjects was defined as participants with a greater SBP than 140 mmHg, a DBP than 90 mmHg, or the use of antihypertensive medication. Abbreviations: OR, odds ratio; CI, confidence interval; BP, blood pressure; HbA1c, hemoglobin A1c; LA, linoleic acid.

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
