# Peer review of "Relationship between Dietary n-6 Fatty Acid Intake and Hypertension: Effect of Glycated Hemoglobin Levels"

_nutrients, 2018, doi:10.3390/nu10121825_

Reviewer 1 Report

This manuscript (ms) describes the association between the nominal dietary intake of linoleic acid (LA) and blood pressure in elderly subjects in Ishikawa Prefecture in Japan. It is a cross sectional study.

There are some issues the authors should consider to improve the ms.

1.    The authors unexpectedly found an association between high HbA1c and subjects with high relative dietary LA and raised BP. In other words, high relative dietary LA was positively associated with raised BP.

2.    It is important that the authors provide additional data on plasma/serum levels of LA, glucose and insulin to see if the story they are telling is supported by additional biochemical indicators.

3.    Plasma LA levels are really important to support the observations of this study given the great uncertainty about dietary data (which the authors acknowledge).

4.    Can the authors provide the readers with proposed mechanism of action for the associations described.

5.    Line 254….states ‘there are few studies that have investigated the relationship between dietary n-6 FA and hypertension by the extent of HbA1c”.  Can the authors provide details of what these few studies found?

6.    It would be of great interest for the authors to provide a 3-dimensional plot of LA, BP and HbA1c to illustrate their results more clearly.

7.    Can the authors explore whether there is a gender effect in their data? 

8.    Many important references are old (1980, 1984, 1993 etc, references 17, 19, 20) and the authors should replace these with more up-to-date papers that describe current theories on the mechanism of action of n-6 fatty acids and blood pressure, and the likely mediators of the effect.

9.    Reference 18 does not appear to be the correct reference (it is a methods paper).

10.Table 1 why don’t the Sat+Mono+PUFA add up to the total Fat?

11. Table 1 – P values are given, but it is not clear which groups are significantly different. Please consider indicating this with a letter or appropriate accepted notation (letters/asterisks).

12. Please indicate which Ethics Committee gave approval for the study.

Author Response

Dear Ms. Kyra Yan and Reviewer 1

We greatly appreciate your review of our manuscript and the provision of helpful suggestions. Below are our responses to the reviewers’ comments, with a description of the changes made to the manuscript. In the revised manuscript, red text indicates the portions revised according to the comments of Reviewer 1.

 Response to Reviewer 1

 We greatly appreciate your helpful comments and suggestions. Changes made in accordance with your comments are indicated by red text in the revised manuscript.

This manuscript (ms) describes the association between the nominal dietary intake of linoleic acid (LA) and blood pressure in elderly subjects in Ishikawa Prefecture in Japan. It is a cross sectional study.

There are some issues the authors should consider to improve the ms.

1.    The authors unexpectedly found an association between high HbA1c and subjects with high relative dietary LA and raised BP. In other words, high relative dietary LA was positively associated with raised BP.

2.    It is important that the authors provide additional data on plasma/serum levels of LA, glucose and insulin to see if the story they are telling is supported by additional biochemical indicators.

3.    Plasma LA levels are really important to support the observations of this study given the great uncertainty about dietary data (which the authors acknowledge).

I’m sorry that we cannot use data on levels of LA, glucose and insulin in this study.

Considering the reviewer’s suggestions, we added the following description to the revised manuscript: “Fourth, dataon serum levels of n-6 fatty acids were not used. N-6 fatty acids are essential fatty acids and humans must intake them because the body cannot synthesize them. With this taken into consideration, serum levels of n-6 fatty acids depend on amount of dietary consumption of n-6 fatty acids. However, serum concentration of n-6 fatty acids is considered to be a better indicator of n-6 fatty acid status and the relationship between serum levels of n-6 fatty acids and hypertension needs to be confirmed in further studies.” In line 317 in the Discussion (Limitation) section. 

We also added the following description to the revised manuscript to explain the validity of BDHQ for n-6 fatty acids: “Regarding n-6 fatty acids, the Pearson correlation coefficients with the dietary records for the energy-adjusted intakes ranged from 0.49 in men and from 0.55 in women” in line 108 in the Material and Methods section. 

 4.    Can the authors provide the readers with proposedmechanism of action for the associations described.

Based on the reviewer’s suggestions, we added the following description to the revised manuscript: “We consider that when dietary n-6 fatty acids are converted into arachidonic acids in vivo, PG vasodilators such as PGI2 and PGE2 are produced from arachidonic acids and then these vasodilator PGs lower BP in general population. On the other hand, more vasoconstriction TXs such as TXA2 are produced from arachidonic acids in diabetic subjects than in non-diabetic subjects, and the elevated TXA2/PGI2 consequently causes hypertension in diabetic subjects. In addition, the elevated activation of TXA2 in diabetic state is considered to cause hypertension.” in line 292 in the Discussion section.

 5.    Line 254….states ‘there are few studies that have investigated the relationship between dietary n-6 FA and hypertension by the extent of HbA1c”.  Can the authors provide details of what these few studies found?

We cannot find reports that demonstrated the relationship between the consumption of dietary n-6 fatty acids and hypertension in diabetic subjects.

We changed description of the corresponding part in the revised manuscript; “To the best of our knowledge, the relationship between the consumption of dietary n-6 fatty acids and hypertension by the extent of a HbA1c level has not yet been studied.” in line 281 in the Discussion section

 6.    It would be of great interest for the authors to provide a 3-dimensional plot of LA, BP and HbA1c to illustrate their results more clearly.

Based on the reviewer’s suggestions, we made a 3-dimensional graph of dietary n-6 fatty acid or LA intake, BP and HbA1c (Figure 2). In this figure, we showed a P value for interactionbetween BP and HbA1c groups by n-6 fatty acid intakecalculated by a two-way ANOVA. 

 7.    Can the authors explore whether there is a gender effect in their data? 

Based on the reviewer’s suggestions, we changed Table 1 to show the characteristics in different according to gender. We also added following description in revised manuscript; “Regarding differences of gender, female had a lower BMI, SBP and DBP and consumed more n-6 fatty acid and LA. There was no significant difference in HbA1c between male and female.” in line 153 in Results section.

In addition, we performed a two-way ANOVA to assess interaction between BP and gender groups by n-6 fatty acid intake(Supplementary Table 2). We added following description in revised manuscript; “Unlike the interaction of HbA1c groups, there were no significant interaction between BP groups and gender groups in consumption of n-6 fatty acids (Supplementary Table 2)” in line 188 in Results section.

 8.    Many important references are old (1980, 1984, 1993 etc, references 17, 19, 20) and the authors should replace these with more up-to-date papers that describe current theories on the mechanism of action of n-6 fatty acids and blood pressure, and the likely mediators of the effect.

 Based on the reviewer’s suggestion, we added and changed up-to-data papers as references.

9.    Reference 18 does not appear to be the correct reference (it is a methods paper).

Based on the reviewer’s indication, we excluded the reference 18.

  10.Table 1 why don’t the Sat+Mono+PUFA add up to the total Fat?

Fat is considered to consist of total fatty acids (SFA + MUFA + PUFA), cholesterol, phospholipid, glycolipid and so on. In this study, we can only use the data on total fat, SFA, MUFA, PUFA and cholesterol. In order to avoid misunderstandings, we exclude the data of fat and cholesterol in Table 1, Table2 and Table 3.

11. Table 1 – P values are given, but it is not clear which groups are significantly different. Please consider indicating this with a letter or appropriate accepted notation (letters/asterisks).

Based on the reviewer’s suggestion, we added explanations for p-value in the footnote in Table 1.

12. Please indicate which Ethics Committee gave approval for the study.

Based on the reviewer’s suggestions, we added the following description to the revised manuscript: “The SHIKA study was approved by the Ethical Committee at Kanazawa University.” in line 146 in the Methods section.

Thank you in advance for considering our revised manuscript for publication in Nutrients.

 Respectfully yours.

 Haruki Nakamura

 Haruki Nakamura, MD.

The Department of Environmental and Preventive Medicine at the Kanazawa University Graduate School of Medicine

13-1 Takara-machi, Kanazawa, Ishikawa, 920-8640, Japan.

Tel.: +81-76-265-2218; Fax: +81-76-234-4233; 

E-mail: haruki_nakamura@stu.kanazawa-u.ac.jp

Reviewer 2 Report

This is a well written and clear manuscript and I have few comments.

Additional information on the sampling strategy and representativeness of the sample from Shika would be helpful. First, the very high response rate (1987 out of 2160 returning questionnaires) is notable, but then only 837 underwent medical exams which is very low. Additional interpretation/comment/details on this would be helpful. 

    2. I found Table 2's p-values to be a bit unclear and/or insufficient to be helpful. The interaction p-values are clear. However, P1 and P2 are unclear. Are these p-values on the marginal means (e.g., overall HbA1c group means compared to each other ignoring BP) - if so, why are these interesting to look at? Furthermore, the marginal means are not reported in the table. Of more/relevant interest I think would be to compare the means in the table directly. For example, 5.59 vs. 4.54 is probably saying that there is a significant difference (what is the p-value??) for the association of HbA1c level with n6 level for those with normal BP, whereas 5.12 vs. 5.11 (what is the p-value) is probably saying that there is not a significance difference/association of HbA1C with n6 for hypertensives. Similar for LA, etc. This information is currently in Table 3, but maybe move it to the footnote on Table 2 as well? 

 3. While table 4 makes sense I'm a bit confused with why the switch to hypertension status (y/n) as the response variable when it's n6 dietary intake in table 2? What is the conceptual model being tested? Would it be cleaner to just start with hypertension as the outcome throughout and then look at how n6 and hba1c interact to inform hypertension status from the outset of the paper and let that framework drive the entire analysis vs. switching things up part way through? 

4. I could have used more on the n-6 overall vs. LA distinction. Is LA part of the total n-6 value? So, that 5.15 (LA mean) < 5.30 (n6 mean) means that you are only estimating and average of 0.15 n6 from other sources? If so, what other n6s are you quantifying from the survey? Finally, did you test to see whether the interaction you observed is also there for the non-LA n6's? Seems helpful to test that and, at least, indicate whether it was or wasn't significant. 

Typo: Table 3, the p-value for n3 fatty acid has a comma instead of a decimal

Author Response

Dear Ms. Kyra Yan and Reviewer 2

We greatly appreciate your review of our manuscript and the provision of helpful suggestions. Below are our responses to the reviewers’ comments, with a description of the changes made to the manuscript. In the revised manuscript, blue text indicates the portions revised according to the comments of Reviewer 2.

 Response to Reviewer 2

 We greatly appreciate your helpful comments and suggestions. Changes made in accordance with your comments are indicated by blue text in the revised manuscript.

This is a well written and clear manuscript and I have few comments.

Additional information on the sampling strategy and representativeness of the sample from Shika would be helpful. First, the very high response rate (1987 out of 2160 returning questionnaires) is notable, but then only 837 underwent medical exams which is very low. Additional interpretation/comment/details on this would be helpful. 

 Based on the reviewer’s suggestion, we added following description in revised manuscript: “Medical examinations were solicited from the 1987 participants and active collaborators were included in this study. Among 1987 participants, 837 participants were voluntary applicants for medical examinations, and 1150 participants did not undergo a medical examination.” in line 78 in the Materials and Methods section.

We also revised the manuscript; “Third, a selection bias and a treatment effect bias needs to be considered. Participants were voluntary collaborators and they might be people who were concerned about being healthy. In addition, patients receiving medication for hypertension may also have been receiving lifestyle guidance from their physician.” in line 314 in the discussion (limitation) section.

    2. I found Table 2's p-values to be a bit unclear and/or insufficient to be helpful. The interaction p-values are clear. However, P1 and P2 are unclear. Are these p-values on the marginal means (e.g., overall HbA1c group means compared to each other ignoring BP) - if so, why are these interesting to look at? Furthermore, the marginal means are not reported in the table. Of more/relevant interest I think would be to compare the means in the table directly. For example, 5.59 vs. 4.54 is probably saying that there is a significant difference (what is the p-value??) for the association of HbA1c level with n6 level for those with normal BP, whereas 5.12 vs. 5.11 (what is the p-value) is probably saying that there is not a significance difference/association of HbA1C with n6 for hypertensives. Similar for LA, etc. This information is currently in Table 3, but maybe move it to the footnote on Table 2 as well?

We performed two-way ANOVA to assess “interactions”. In order to avoid misleading, we excluded P1 and P2 in Supplementary Table 1 and Supplementary Table 2. The differences of n-6 fatty acids intake between Hypertension group and Normal BP group according to HbAc1 level were shown in Table 3.

3. While table 4 makes sense I'm a bit confused with why the switch to hypertension status (y/n) as the response variable when it's n6 dietary intake in table 2? What is the conceptual model being tested? Would it be cleaner to just start with hypertension as the outcome throughout and then look at how n6 and hba1c interact to inform hypertension status from the outset of the paper and let that framework drive the entire analysis vs. switching things up part way through?

A two-way ANOVA was performed to demonstrate the interaction between BP groups and HbA1c groups. Since we defined hypertension as a “qualitative variable” in the present study and used hypertension as an independent variable in the two-way ANOVA, the analysis demonstrated an interaction between BP groups and HbA1c groups. The significance of the interaction provided information on the difference in the intake of n-6 fatty acid intake between BP groups and HbA1c groups and, therefore, we considered the interaction to provide supporting evidence of the stratification by HbA1c level to perform a separate analysis on high HbA1c and normal HbA1c subjects in a multiple logistic analysis (Table 4).

To use hypertension as dependent variable, we had to change the definition of hypertension in this study; we needed to exclude subjects with antihypertensive medications. In addition, to use data on n-6 fatty acid as independent variable, we needed to them as qualitative variable. However, the cut-off of n-6 fatty acid intake was not clearly defined. We judged that it was better to use the data on n-6 fatty acid as continuous variable.

4. I could have used more on the n-6 overall vs. LA distinction. Is LA part of the total n-6 value? So, that 5.15 (LA mean) < 5.30 (n6 mean) means that you are only estimating and average of 0.15 n6 from other sources? If so, what other n6s are you quantifying from the survey? Finally, did you test to see whether the interaction you observed is also there for the non-LA n6's? Seems helpful to test that and, at least, indicate whether it was or wasn't significant.

Based on the reviewer’s suggestion, we added the data on non-LA such as gamma-linolenic acid (GLA), eicosadienoic acid (EA), dihomo-gamma-linolenic acid (DGLA), arachidonic acid (AA) and docosapentaenoic acid (DPA) in Table 1, Table 2 and Table 3. We also showed interactions of them (Supplementary Table 1).

In addition, we added following description in revised manuscript; “For other n-6 fatty acids such as such as gamma-linolenic acid, eicosadienoic acid, dihomo-gamma-linolenic acid, arachidonic acid and docosapentaenoic acid, no significant interactions were observed (Supplementary Table 1)in line 186 in the Results section

Typo: Table 3, the p-value for n3 fatty acid has a comma instead of a decimal.

Based on the reviewer’s indication, we revised the corresponding part.

Thank you in advance for considering our revised manuscript for publication in Nutrients.

 Respectfully yours.

 Haruki Nakamura

Haruki Nakamura, MD.

The Department of Environmental and Preventive Medicine at the Kanazawa University Graduate School of Medicine

13-1 Takara-machi, Kanazawa, Ishikawa, 920-8640, Japan.

Tel.: +81-76-265-2218; Fax: +81-76-234-4233; 

E-mail: haruki_nakamura@stu.kanazawa-u.ac.jp

Round  2

Reviewer 1 Report

The authors have responded well to the advice of the reviews (reviewer 1).

However on re-reading the manuscript it is clear there is still room for significant improvement to provide clarity on the issues detailed below:

Throughout, pay attention to use of the word "hypertension".  In some instances, use hypertensive subjects or subjects with hypertension, rather than hypertension subjects.

L155-…female subjects.

L166 —delete ..’were hypertension..’

L167 …Ratios (not ratio)

Figure 2 is very difficult to interpret and has not indicated significant differences. It is suggested this Figure does not illustrate the data effectively. For example, the Abstract (lines 22-26) state that (1) high n-6 intake was inversely correlated with hypertensive subjects with a low HbA1c, and (2) in subjects with a high HbA1c n-6 intake was significantly associated with hypertension. The Figure should demonstrate this clearly. Currently, it does not.

It is suggested re-draw the Figure. It is important to remember that HbA1c is a continuous variable so readers should be able to see the spread of the relationship between dietary LA and blood levels of HbA1c.  One possibility is to plot the Figure as 2A and 2B as follows. Figure 2A = Subjects with normal blood pressure, and Figure 2B subjects with elevated blood pressure. The Y axis in each case is the dietary intake of LA, and x axis is HbA1c level. On each the cut-off for what is regarded as normal HbA1c can be indicated say as a vertical dotted line.  Such a plot should clearly demonstrate the central novel feature of this manuscript.

Lines 169 and 204 seem to contradict each other! L169 says higher LA intake in normotensive subjects tham hypertensive subjects, while line 204 says the opposite - hypertensive consumed more LA than normotensives! Table 3 for low HbA1c indicate LA intakes were 4.96 (Hyper) versus 5.21 (normo), p<0.001!  Please correct this section of the manuscript as it is very confusing.

Please indicate which are the up-to-date references you have  included and the line numbers where these have been used.

Author Response

Dear Ms. Kyra Yan and Reviewer 1

 We greatly appreciate your review of our manuscript and the provision of helpful suggestions. Below are our responses to the reviewers’ comments, with a description of the changes made to the manuscript. In the revised manuscript, red text indicates the portions revised according to the comments of Reviewer 1.

Response to Reviewer 1

 We greatly appreciate your helpful comments and suggestions. Changes made in accordance with your comments are indicated by red text in the revised manuscript.

Throughout, pay attention to use of the word "hypertension".  In some instances, use hypertensive subjects or subjects with hypertension, rather than hypertension subjects.

L155-…female subjects.

L166 —delete ..’were hypertension..’

L167 …Ratios (not ratio)

 Based on reviewer’s indication, we corrected the points that were pointed out. Descriptions of line 138, 160, 168, 175, 202, 209, 219, 230 and 250 were corrected places in the revised manuscript.

Figure 2 is very difficult to interpret and has not indicated significant differences. It is suggested this Figure does not illustrate the data effectively. For example, the Abstract (lines 22-26) state that (1) high n-6 intake was inversely correlated with hypertensive subjects with a low HbA1c, and (2) in subjects with a high HbA1c n-6 intake was significantly associated with hypertension. The Figure should demonstrate this clearly. Currently, it does not.

It is suggested re-draw the Figure. It is important to remember that HbA1c is a continuous variable so readers should be able to see the spread of the relationship between dietary LA and blood levels of HbA1c.  One possibility is to plot the Figure as 2A and 2B as follows. Figure 2A = Subjects with normal blood pressure, and Figure 2B subjects with elevated blood pressure. The Y axis in each case is the dietary intake of LA, and x axis is HbA1c level. On each the cut-off for what is regarded as normal HbA1c can be indicated say as a vertical dotted line. Such a plot should clearly demonstrate the central novel feature of this manuscript.

 Based on the reviewer’s suggestion, we changed Figure 2 and added following description to the revised manuscript; “Figure 2 demonstrated the spread of relationship between dietary LA intake and blood levels of HbA1c. In subjects with normal HbA1c levels, figure 2A and 2B suggested that the number of subjects consuming high LA was more in normal BP subjects than in hypertensive subjects. In subjects with high HbA1c level, there were tendency for hypertensive subjects to consume high intake of LA, compared with normal BP subjects.” line 183 in Results section.

 In addition, we changed Supplementary Table 1 to Table 3 to show the interaction between BP groups and HbA1c group on n-6 fatty acids.

Lines 169 and 204 seem to contradict each other! L169 says higher LA intake in normotensive subjects tham hypertensive subjects, while line 204 says the opposite - hypertensive consumed more LA than normotensives! Table 3 for low HbA1c indicate LA intakes were 4.96 (Hyper) versus 5.21 (normo), p<0.001! Please correct this section of the manuscript as it is very confusing.

 Based on reviewer’s indication, we changed the description in the revised manuscript; “There was a significant difference in n-6 fatty acids and LA intake among BP groups; normal BP group consumed more n-6 fatty acids (P < 0.001) and LA (P < 0.001) than hypertension group.” line 221 in Results section.

Please indicate which are the up-to-date references you have included and the line numbers where these have been used.

 Articles such as 17, 18, 19, 20, 21, 24 and 25 were up-to-date references. Descriptions of line 48, 53, 54, 304 and 307 were corrected places in the revised manuscript.

Thank you in advance for considering our revised manuscript for publication in Nutrients.

  Respectfully yours.

Haruki Nakamura, MD.

The Department of Environmental and Preventive Medicine at the Kanazawa University Graduate School of Medicine

13-1 Takara-machi, Kanazawa, Ishikawa, 920-8640, Japan.

Tel.: +81-76-265-2218; Fax: +81-76-234-4233; 

E-mail: haruki_nakamura@stu.kanazawa-u.ac.jp
